# 2D Axisymmetric Modeling of Refill Friction Stir Spot Welding and Experimental Validation

Evan Berger, Michael Miles, Andrew Curtis, Paul Blackhurst and Yuri Hovanski *

Department of Manufacturing Engineering, Brigham Young University, Provo, UT 84602, USA
* Correspondence: yuri.hovanski@byu.edu

**Abstract:** The development of the simulation of refill friction stir spot welding (RFSSW) is critical to be able to predict the behavior of aluminum in the process under specific parameters. A two-dimensional axisymmetric thermo-mechanical model of the RFSSW process for 7075-T6 aluminum alloy sheet was developed and validated with experimental data. Welding temperatures and material flow, including defect formation, were accurately predicted by the model. While these results are encouraging, further development of bonding criteria is needed for simulation models, in order to enable the prediction of properties such as joint strength. The simulation was validated by a comparison of temperatures measured in the weld, which were demonstrated to be accurate at all positions in and around the weld nugget, within 10% of measured values. Additional validation of material flow was performed with post-weld optical microscopy where the simulation is shown to be able to predict the presence or absence of internal volumetric defects based on the variation in process parameters. Finally, the prediction of the tool process forces during the welding cycle were evaluated; however, both probe and shoulder forces were overestimated using the standard flow stress data for AA 7075-T6.

**Keywords:** refill friction stir spot welding; model; thermal comparison; microscopy

## 1. Introduction

Friction stir welding (FSW) and friction stir spot welding (FSSW) are solid-state joining processes with increased application in aluminum alloys for automotive, rail, and aerospace industries [1,2]. Currently, the aerospace industry primarily uses riveting for fastening to meet basic requirements and mechanical properties [3,4]; however, solid-state joining processes can eliminate mass addition while simultaneously improving properties [5–7]. Refill friction stir spot welding (RFSSW) is a variant of FSW that was developed to create a flush surface on the top of the weld and to improve joint strength compared to FSSW [8–10]. Preliminary results suggest that RFSSW is capable of meeting strength requirements for many applications, but the process parameter development is often Edisonian at best, with development largely being carried out by trial and error.

Ideally, process parameters would be defined using numerical tools, yet, at present, simulation models have not been shown to be sufficiently predictive for parameter development. Initial numerical simulation of RFSSW started as early as 2006 [11], evaluating the initial deformation induced from the pin coming into contact with the workpiece. Similar evaluations continued until around 2015 [12–14] with different approaches focused primarily on coupled Eulerian–Lagrangian (CEL) formulations.

Between 2010 and 2013, a major shift in the understanding of RFSSW changed the overall tool motion, which originally focused on the pin moving downward into the workpiece, similar to traditional FSSW [15]. However, by 2013, research and development moved to a shoulder first tool motion as shown in Figure 1. As this change in motion to a shoulder first plunge process greatly diverged from more traditional FSW and FSSW modeling techniques, little was reported on simulation tools for RFSSW until more recently, while a body of knowledge related to parameters, properties, and microstructure was disseminated.

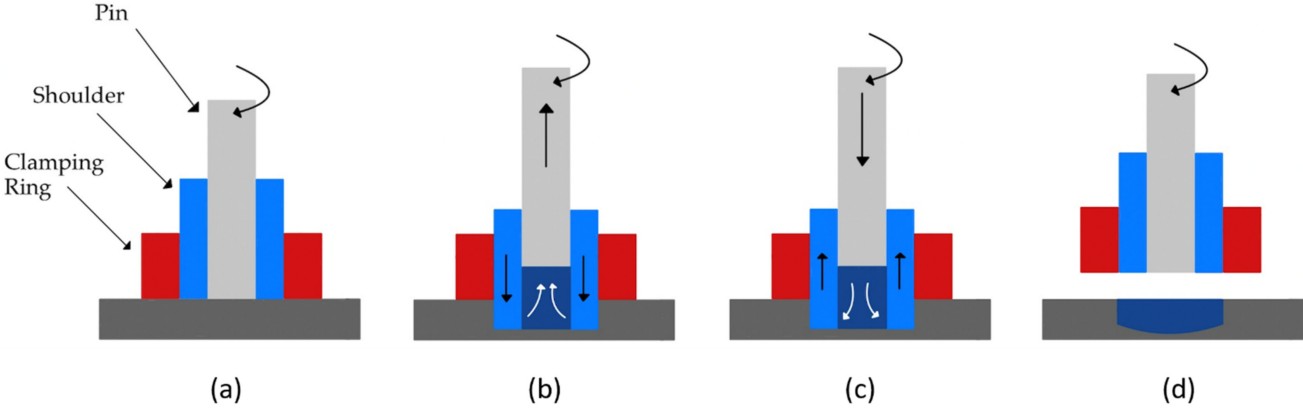

**Figure 1.** Schematic of the RFSSW process: (**a**) clamping phase, (**b**) plunging phase, (**c**) refill phase, and (**d**) retraction phase.

Since 2020, several papers have once again started to focus on numerical tools specific to RFSSW. Kubit et al. demonstrated that a 2D axisymmetric simulation using Simufact Forming Software could be used to represent the interaction of the RFSSW tool with the workpiece [16]. Their results showed an ability to simulate the temperature in both the tool and workpiece with validated results within 14% and 10%, respectively. Furthermore, they showed the ability of the simulation to model incomplete refill at the top of the sheet.

Other publications in 2021 focused on CEL-type formulations in Abaqus/Explicit [17,18], which modeled the effect of the process on the workpiece rather than a coupled model analyzing both the tool and workpiece as well as their interactions. Both groups focused on evaluating temperature and material flow. Xiong et al. evaluated temperature generation over a range of rotational velocities, and concluded that higher temperatures associated with increasing rotational velocities led to asperities in the RFSSW process that resulted in voids. Their work thoroughly investigates the process numerically and relied on experimental correlations of the Zener–Holloman parameter to extrapolate the simulated thermal data. Zhang et al. evaluated the RFSSW of magnesium AZ91D, demonstrating similar results to others, showing that material flow and temperature were simulated.

With growing reports on simulation tools related to RFSSW showcasing an increasing ability to model both flow and temperature generation, the purpose of this study is to further evaluate numerical tools to aid in the evaluation of process parameters. Previous studies focused on the variation in rotational velocities [17] with fixed tool kinematics. However, small variations in tool motion have been shown to be an effective method for eliminating internal defects. As such, this study evaluates the ability of numerical simulation to model the effects of varied tool motions on void elimination.

## 2. Materials and Methods

RFSSW experiments were performed with similar material sheets of 1.6 mm AA7075-T6. The chemical composition of AA7075 is 5.6% zinc, 2% magnesium, 1.5% copper, and less than 0.5% of silicon, iron, and manganese. These composition values are nominal. The T6 heat treatment is a two-step process of quenching and aging to increase the yield strength of the alloy to 476 MPa and the UTS to 538 MPa [19]. While the flow stress data for AA7075-T6 is not readily available for the range of temperatures and strain rates appliable to the RFSSW process, one of the aims of this work was to determine the influence of incoming material properties on the validity of a predictive tool.

The tooling setup for the RFSSW process includes a pin, shoulder, and rigid clamp, as seen in Figure 2. The clamp, shoulder, and pin are all concentric and allow for kinematic motion of the rotating pin and shoulder with the non-rotating clamp such as the movement depicted in Figure 1. All experimentation included in this study enabled the rotational motion of pin and shoulder to spin at 2600 rpm in the same direction. These parameters were developed and documented previously as a means of significantly reducing the

process cycle time of RFSSW [20,21]. The four-step process highlighted in Figure 1 starts with the non-rotating clamp descending until the rotating pin/shoulder combination comes in contact with the surface of the upper sheet. The second step, Figure 1b, allows the rotating shoulder to plunge into the workpiece, while the rotating pin rises into the tool set, creating a cavity for the material being displaced by the shoulder. The third step, Figure 1c, returns the shoulder and probe to the surface of the material such that the displaced material is pushed back into the workpiece as the shoulder is raised from the maximum plunge depth it achieves during the second step. The final step, Figure 1d, shows the tooling pulling away from the workpiece with a flush weld completed.

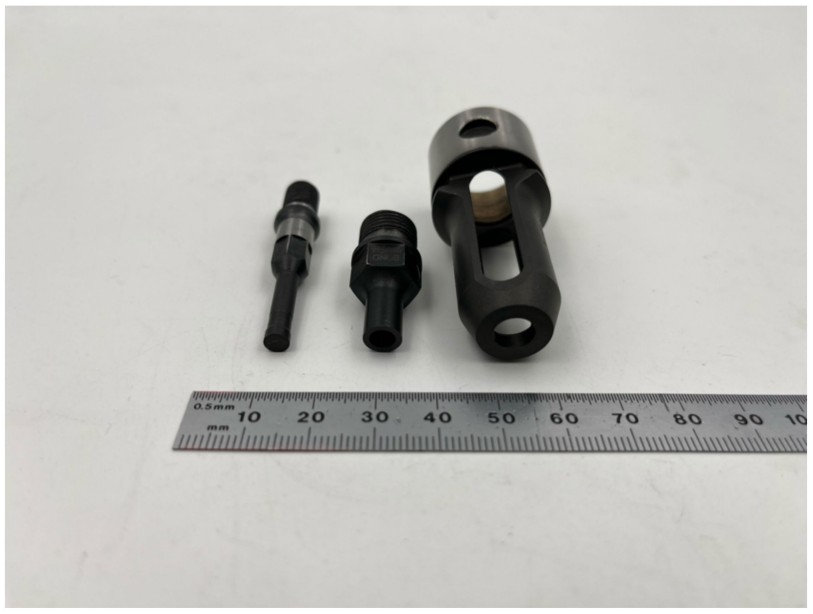

**Figure 2.** RFSSW tooling, including a pin (**left**), shoulder (**center**), and clamping ring (**right**).

The experiment is completed in two modes: (1) volumetric and (2) non-volumetric. The volumetric kinematics are symmetric; raising the pin and plunging the shoulder into the material at steady rates that represent equal volumes of material displaced, until the shoulder is fully plunged. In this symmetric mode, the shoulder and pin are then returned to their original positions in the same manner, volumetrically constrained to the motion of shoulder and pin. Volumetric kinematics are the traditional means that researchers have used to control RFSSW for more than a decade. However, recent updates in equipment capabilities designed to make the RFSSW more amenable to high volume production applications have enabled significant reductions in cycle times [20,21], which have necessitated changes to the traditional volumetric tool kinematics in order to enable cycle times near or below one second.

The non-volumetric kinematics are not symmetric. In this mode, the shoulder is plunged the first 1.5 mm based on volumetric motion of the pin, but during the last 0.5 mm of motion the pin motion is reduced such that pressure is built against the pin as the shoulder plunges more than the volumetric equivalent of the pin. Similar to this non-volumetric plunge of the shoulder, returning the shoulder and pin to the surface is carried out in such a way as to avoid a change to a completely hydrostatic pressure condition within the weld material. By using non-volumetric kinematics, cycle times have been successfully demonstrated for AA7075-T6 as low as 600 ms, while simultaneously showing an increase in as-welded properties [20]. These studies refuted the previously documented understanding that weld cycle times below 2 s only led to reduced properties of RFSSWs.

As the specific geometries of the tool are critical to calculating the motion and understanding the influence of the two modes of kinematic motion described herein, the geometries of the tools used for the experiments were replicated exactly in the model,

as any small deviations in these designs could result in significant changes in material deformation, heat generation, and tooling forces in the model prediction. The tools are made of H13 steel, and the key dimensions are shown in Table 1.

**Table 1.** Tool dimensions used for numerical model and validation.

| Tool | Dimension (mm) |
| --- | --- |
| Pin outer diameter | 4.40 |
| Shoulder inner diameter | 4.45 |
| Shoulder outer diameter | 7.00 |
| Rigid clamp inner diameter | 7.05 |
| Rigid clamp outer diameter | 15.00 |

The equipment used for the production of RFSSW and data acquisition throughout the experiments was a Bond Technologies RFSSW end-effector, Figure 3. The machine outputs the positions of the tools and the forces on the tools at any given point in time. The kinematics of the shoulder tool are output directly, where the initial position of the tool (0 mm displacement) corresponds to the tool resting on top of the material. The kinematics follow as the shoulder plunges into the material and returns to the material surface. These positions are negative because the shoulder travels below the reference position. The kinematics of the pin, on the other hand, are programmed in reference to the kinematics of the shoulder, which aligns with Figure 4.

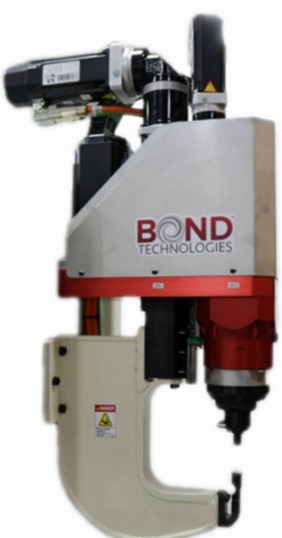

**Figure 3.** Bond Technologies RFSSW robotic end-effector.

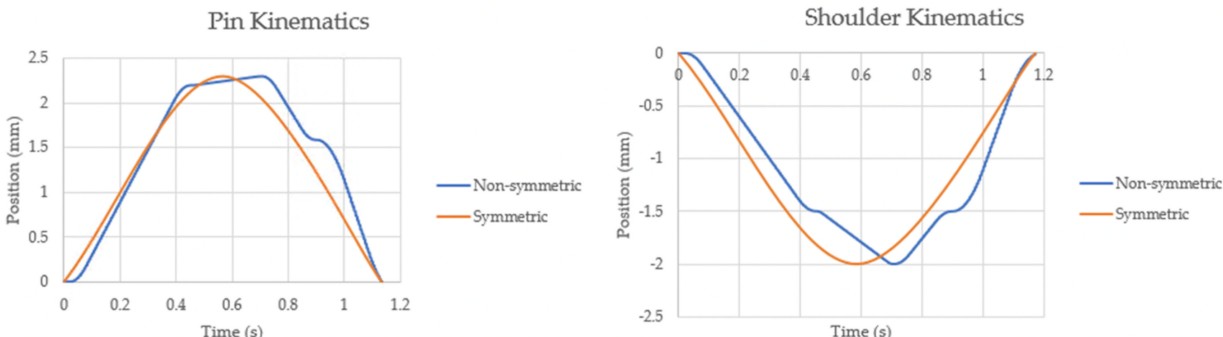

**Figure 4.** Visualization of the kinematics for the tooling in the welding process. Kinematics of the pin. Kinematics of the shoulder.

To measure the temperatures during the welding process, five K-type thermocouples were embedded into the lower of the two sheets of AA 7075-T6, similar to that shown in prior work [16]. This was carried out by milling out paths for the thermocouple wires, as shown in Figure 5, attaching the thermocouples in precise locations in the sheet, and clamping the two sheets of aluminum together. The upper of the two sheets is unaltered.

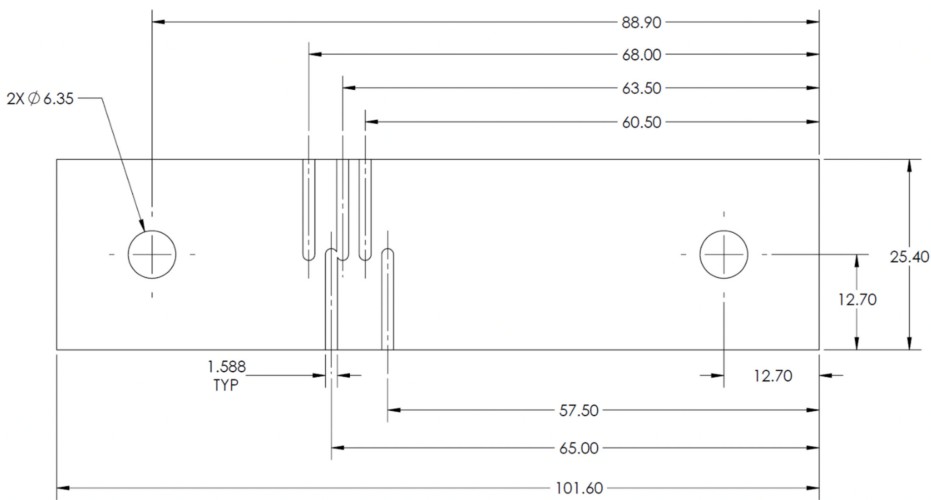

**Figure 5.** Thermocouple layout for CNC mill of AA 7075-T6 coupon.

The thermocouples are labeled 1–5, from the center of the weld to the outside of the stir zone. In the model, five numerical sensors are labeled 1–5, as shown in Figure 6b, which are in the same locations as those in the experiment, shown in Figure 6a. A section view of the thermocouples and sensors is shown in Figure 6b. Both the thermocouples and the sensors are below the stir zone and remain undisturbed during the welding process.

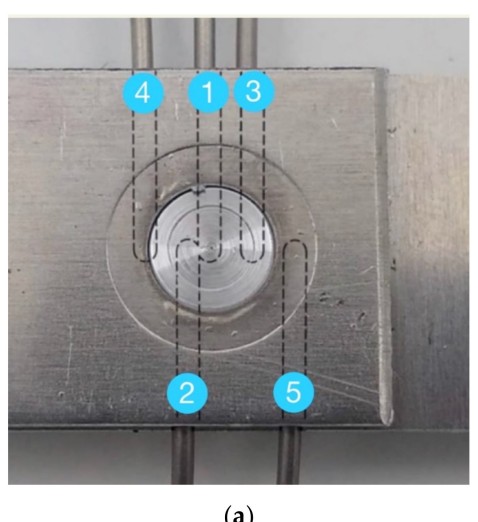

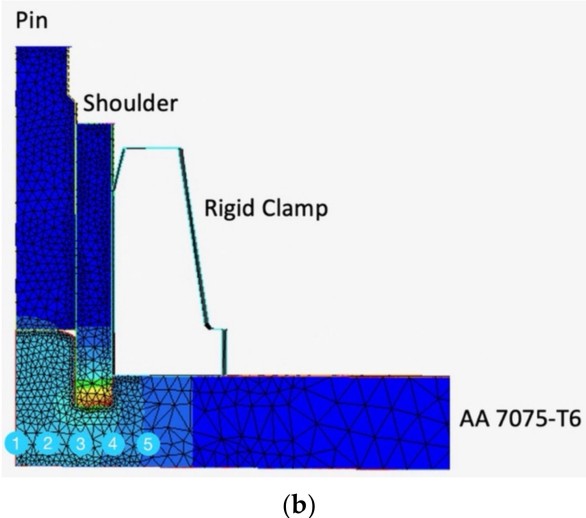

(**a**)                             (**b**)

**Figure 6.** Thermocouple layout for the (**a**) welding process and the (**b**) model.

### 3. Simulation of RFSSW

The finite element (FE) modeling software used to simulate the RFSSW process is ForgeNxt 3.2 developed by Transvalor. Because the RFSSW tooling has rotational symmetry, a 2D axisymmetric approach was used for modeling, as shown in Figure 6b, with a Lagrangian description of material flow to manage the non-steady state nature of the process.



An isotropic, viscoplastic Norton–Hoff law was used to model the flow stresses of the AA7075-T6 material, with the expression for the deviatoric stress tensor, *s*, shown below:

$$s = 2K\left(\sqrt{3}\dot{\bar{\varepsilon}}\right)^{m-1}\dot{\varepsilon} \tag{1}$$

where $\dot{\varepsilon}$ is the strain rate tensor, $\dot{\bar{\varepsilon}}$ is the effective strain rate, $K$ is the material consistency, and $m$ is the strain rate sensitivity of the material. The material consistency $K$ is a function of a strength coefficient $K_o$, temperature $T$ and equivalent strain $\bar{\varepsilon}$, with strain hardening coefficient n and thermal parameter $\beta$, as seen in Equation (2):

$$K = K_0(\varepsilon_0 + \bar{\varepsilon})^n e^{\frac{\beta}{T}} \tag{2}$$

This constitutive equation was fit to data for AA7075-T6, for a variety of different strain rates and temperatures representative of RFSSW, using a standard data set provided in the ForgeNxT 3.2 material database. Friction was modeled using a viscoplastic Norton law, as shown below:

$$\tau_f = -\alpha_f K(T, \bar{\varepsilon}) \| \Delta v_s \|^{q-1} \Delta v_s \tag{3}$$

where $\alpha_f$ is a friction coefficient, $q$ is sensitivity to the relative sliding velocity at the tool/workpiece interface, $\Delta v_s$ is the relative sliding velocity between the tool and the workpiece, and $K$ is a hardening parameter, as previously defined in Equation (2). Therefore, the shear stress at the interface is a function of the local material hardness of the workpiece (the softer material) and the relative sliding velocity. Because the sliding velocity is rate sensitive, greater velocities increase the frictional shear stress, all other things equal.

Calculation of the flow of material is based on a finite element discretization using an enhanced (P1+/P1) 3-noded triangular element, where equilibrium equations are solved at each increment using the Newton–Raphson method. Three zones of higher mesh density are added to the model in the stir zone, where the material is being deformed, to allow the model to more accurately calculate the heat transfer and material flow. These zones were refined during model development, in order to ensure there were enough nodes on the surface of the sheet to follow the tool contours, while also having higher densities in areas where large thermal and deformation gradients were present. The unilateral contact condition is applied to the material surfaces by means of a nodal penalty formulation, where the RFSSW tools and backing plate are considered rigid. An explicit time integration scheme was used to update the sheet geometry at each increment of calculation:

$$X_{t+\Delta t} = X_t + V_{mesh}\Delta t \tag{4}$$

where $X$ is the mesh material coordinate, $V_{mesh}$ is the velocity of the mesh at time $t$, and $\Delta t$ is a time increment chosen sufficiently small. While the tool and backing plate are considered to be mechanically rigid, the evolution of temperature in the tool was modeled in order to provide accurate boundary conditions at the tool/sheet interface. The temperature in both the RFSSW tools and the sheet were calculated at the end of each material flow increment. The two sheets of material are modeled as one thick sheet of material, in the overlap area, for the simplification of the model, from a contact perspective The calculated velocity field allowed the computation of strain rates and stresses in the sheet, which were then used to determine the heat dissipated by friction and plastic deformation. The heat dissipated by plastic deformation (derived from the velocity field) is given by the following term:

$$\dot{q}_v = f\bar{\sigma}\dot{\bar{\varepsilon}} \tag{5}$$

where $\bar{\sigma} = \sqrt{\frac{3}{2}s : s}$ is the equivalent stress (where $s$ is the stress deviator tensor from Equation (1)). The factor f considers the fraction of deformation energy converted into heat,

taken as 0.9 in this paper. For a Norton–Hoff viscoplastic material, the heat generation rate from material deformation is therefore determined as follows:

$$\dot{q}_v = fK\left(\sqrt{3}\dot{\bar{\varepsilon}}\right)^{m+1} \tag{6}$$

An energy balance at the interface between the sheet and the tool yields the following relationship:

$$-k\frac{\partial T}{\partial n} = -h_c(T + T_{tool}) + \frac{b}{b + b_{tool}}\tau \cdot v_s \tag{7}$$

where $T$ is temperature on the surface of the sheet, $T_{tool}$ is the temperature imposed by the tool in contact with the sheet, $n$ is the unit vector normal to the tool surface, and $h_c$ is the conduction heat transfer coefficient. The terms $b$ and $b_{tool}$ are effusivities of the sheet and the tool, respectively. Effusivity is defined as $\sqrt{\rho c k}$, where $\rho$ is density, $c$ is heat capacity, and $k$ is conductivity. The shear stress, $\tau$, is provided by the friction law in Equation (3) and $v_s$. is the relative sliding velocity of the sheet on the tool. The heat dissipated by sliding friction at the tool/sheet interface is shared unequally in this case, where the material with higher effusivity receives a greater portion of the frictional heat. The thermal conductivity, heat capacity, and density of both the H13 tool and the AA 7075-T6 sheet were modeled as a function of temperature, over the range of temperatures from 25 °C to 550 °C. Figure 7 shows the stress–strain curves for various strain rates at 450 °C in AA 7075-T6. Flow stress data across the range of temperatures from 25–550 °C were used to model material behavior, at strain rates up to 100 s$^{-1}$ and for plastic strains up to 4.

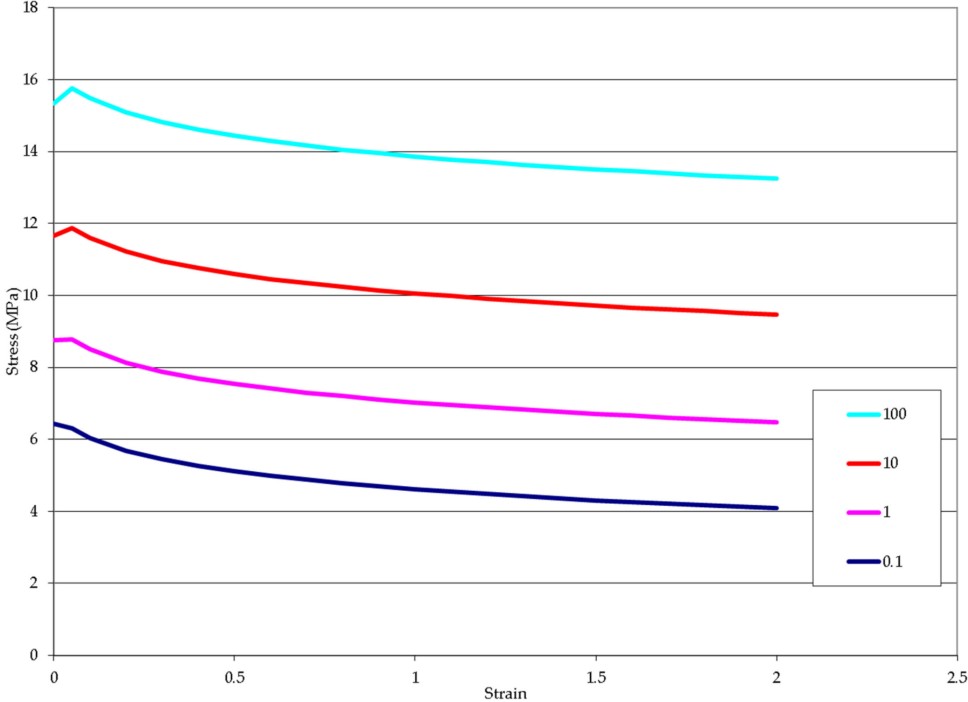

**Figure 7.** Strain rate–sensitive stress curves for AA 7075-T6 at 450 °C.

Finally, the tools were animated with rigid contours, which provided velocity boundary conditions according to the kinematics that were programmed into the code. These contours also imposed an adiabatic thermal boundary condition, as did the rigid contour that provided support to the sheets. The adiabatic boundary conditions were employed, because the welding cycle time is short and therefore heat transfer between the backing plate and the sheets, or between the tool holders and the tools, would be negligible.

## 4. Results

### 4.1. Temperature Predictions

The temperature of the aluminum during the welding process is critical to the success of the weld. If the temperatures are not right, the material flow will be disrupted, and the weld will have neither the required strength nor the desired surface finish. The prediction of the temperature of the aluminum is even more critical as it demonstrates that the heat generation has been properly modeled. The temperatures from the welding process are presented in Figures 8–12, and are compared to those predicted by the model, as seen in Figure 13.

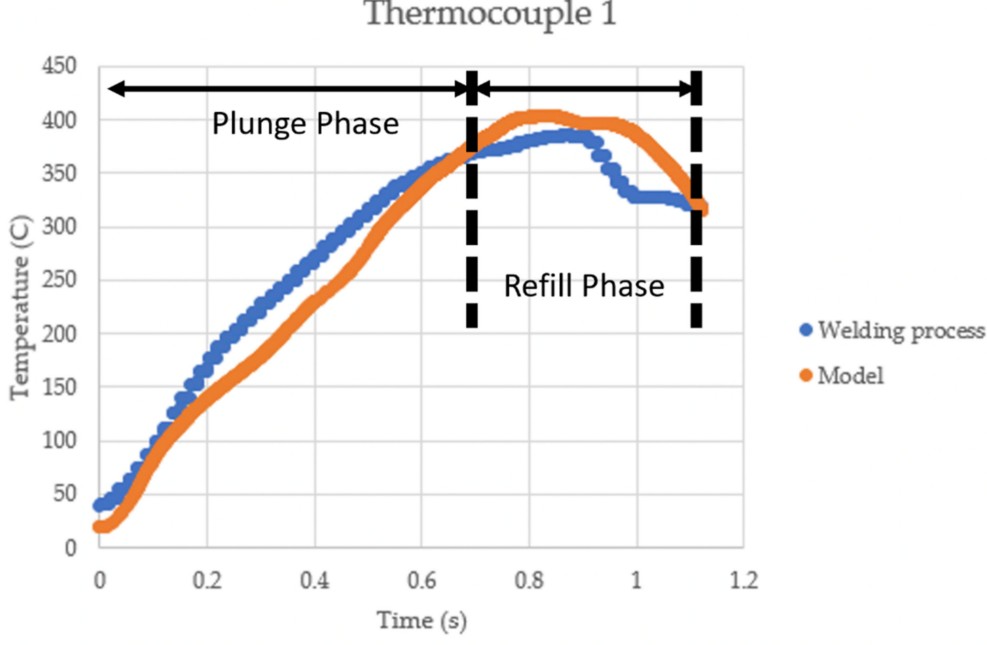

**Figure 8.** Temperature comparison for Thermocouple 1.

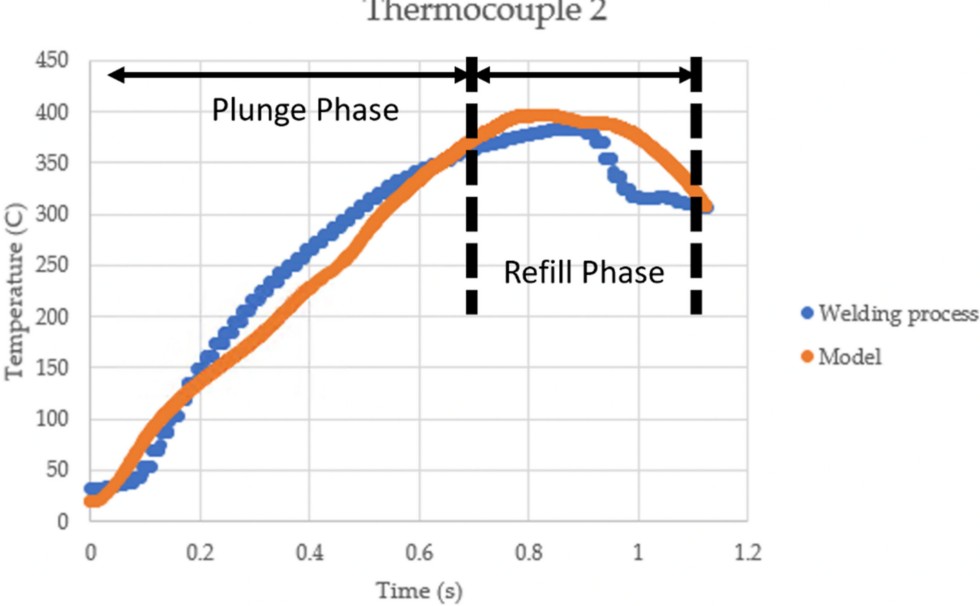

**Figure 9.** Thermal comparison for Thermocouple 2.

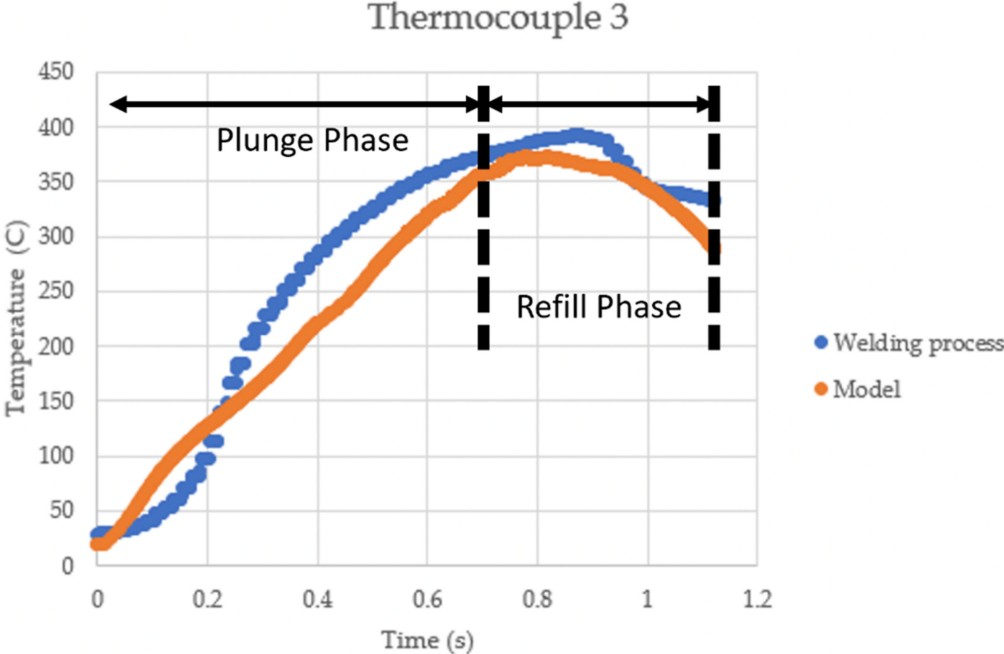

**Figure 10.** Thermal comparison for Thermocouple 3.

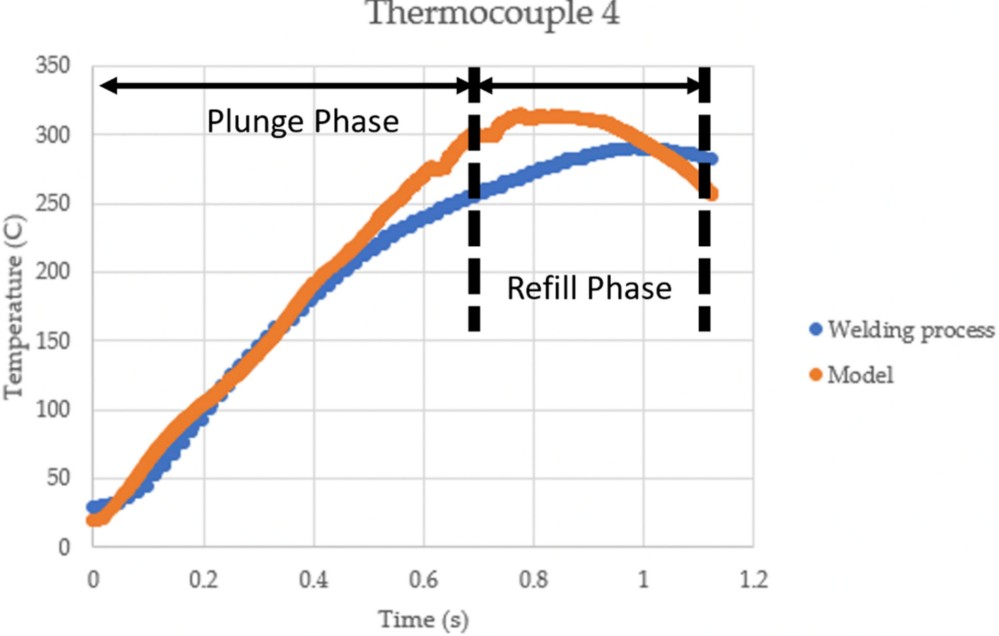

**Figure 11.** Thermal comparison for Thermocouple 4.

The temperature comparison between the experimental validation and the simulation differs by 12% at most for the entire duration of the weld, with most of the predicted temperatures well within 10% of the experiment. The accuracy of the predictions indicates that the heat generation and material flow in the model resemble the welding process. Temperature peaks for the five thermocouples and numerical sensors within the model are displayed in Figure 13. This comparison shows that peak temperatures at each location were within 10% of the measured values.

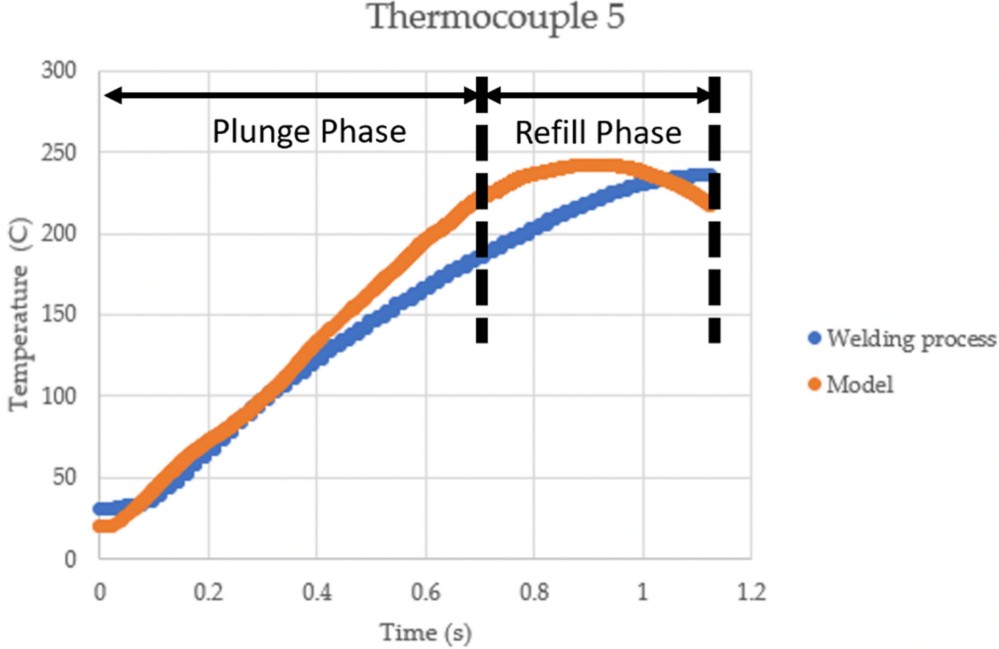

**Figure 12.** Thermal comparison for Thermocouple 5.

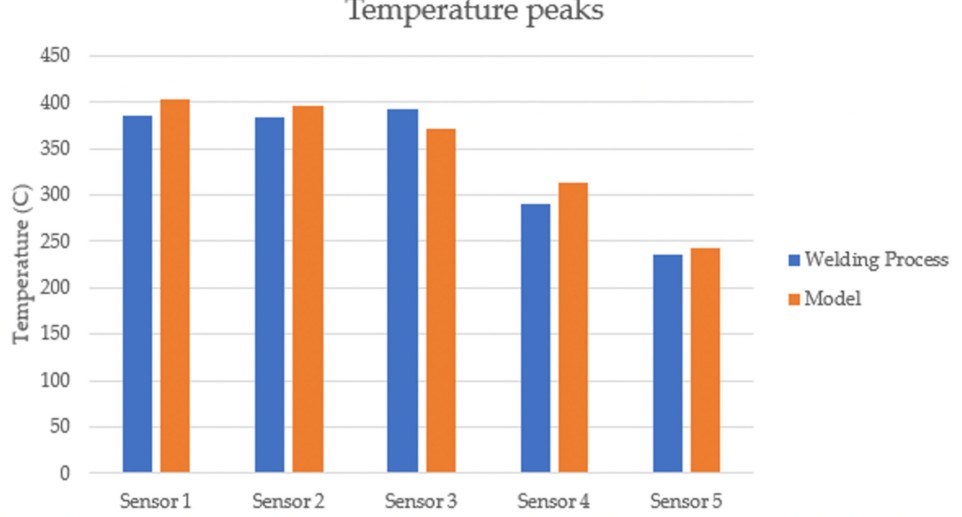

**Figure 13.** Visualization of the temperature peaks for each thermocouple.

These temperature predictions give confidence that the material flow generated by the model is reasonably accurate. The model demonstrates that there is heat generation in the material from both friction and material deformation, though frictional heating dominates for the RFSSW process. Figure 14b–d show the temperature evolution that occurs as the material is heated by the rapid rotation of the tools and consolidated by the relative motion of the pin and shoulder. The heat generation in the model is consistent with the experimental data, which is important if the material flow is to be simulated correctly.

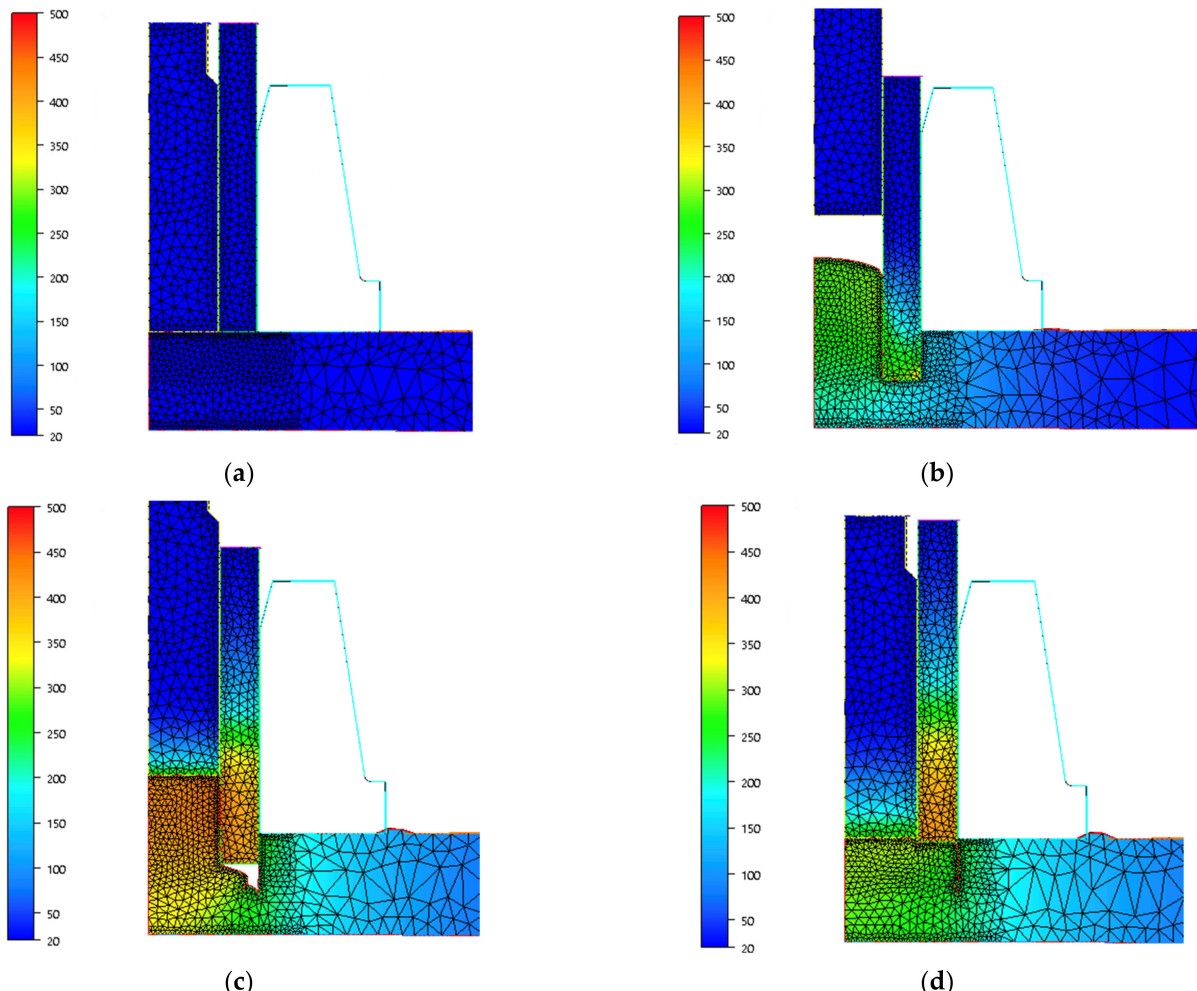

**Figure 14.** Visualization of the heat generation in the model: (**a**) initial position, (**b**) plunge stage at t = 0.5 s, (**c**) refill stage at t = 1.0 s, and (**d**) final position.

### 4.2. Material Flow and Defect Formation

Further validation of the model is given by the observation material flow under the two conditions introduced earlier, namely, volumetric and non-volumetric tool kinematic motion. Under traditional volumetric kinematic motion of the tool shoulder and probe, a void formed in the material at the lower-outside corner of the path of the shoulder. This void is only seen in volumetric welds or welds with symmetric kinematics [22]. Cross-sections were taken from the weld specimens and prepared for optical microscopy.

Figure 15 shows a comparison of the simulated results and the weld specimen with a void having a similar shape (though the model void is larger), in approximately the same location along the boundary where material is folded together during the final stage of welding (this interface is shown in red in Figure 15a). In the model, the interface created by material folding onto itself is maintained by the same contact algorithm that manages nodal contact with the rigid tool. So, during the remeshing of the workpiece, the interface remains and allows the prediction of material flow phenomena, such as the formation of a void seen in Figure 15b.

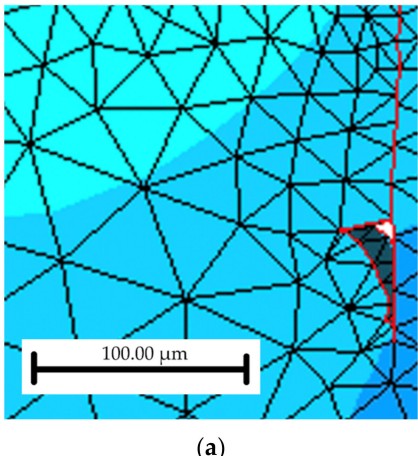

(**a**)

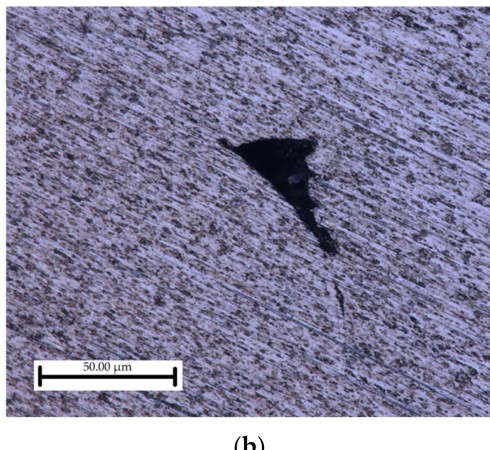

(**b**)

**Figure 15.** A visualization of the void in the material caused by volumetric kinematics. (**a**) The void in the model; (**b**) The void in the welding process.

The red lines in the model indicate the boundary where the materials were folded together during the weld. As such, the triangular area with the red outline is the void where the material did not bond together. Unlike the volumetric case, voids were not found in non-volumetric welds with non-symmetric kinematics. Figure 16 shows the simulated and experimental results, where proper consolidation took place during the joining process. Figure 3 shows how the pin spends a longer time refilling the weld at a more constant rate with non-symmetric kinematics, rather than refilling at the same rate the shoulder plunges. The motivation for non-volumetric tool motion was the proper consolidation of the material. While these weld parameters had previously been documented [20], the modeled material flow herein showing the ability to eliminate the void using these conditions is encouraging. As weld cycle times in 7075-T6 at or below 1 s have only recently been studied, the ability to demonstrate the elimination of internal voids with parameter optimization at these reduced cycle times in the RFSSW is novel.

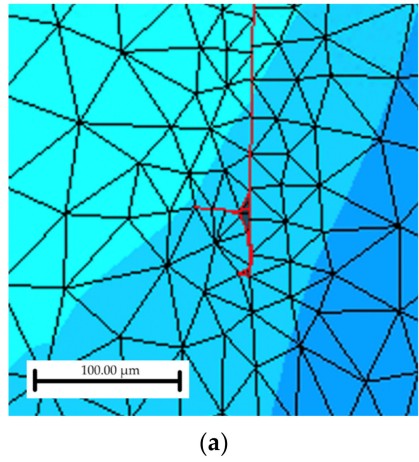

(**a**)

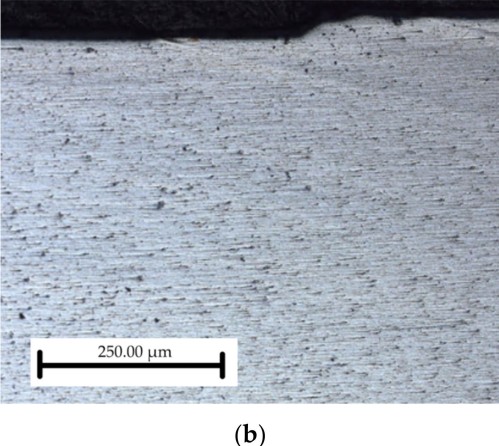

(**b**)

**Figure 16.** A visualization of the lack of a void in the material caused by non-volumetric kinematics. (**a**) The lack of a void in the model; (**b**) The lack of a void in the welding process.

### 4.3. Forces

The forces in the tooling are another indicator of the validity of the model. Correct forces indicate that the heat generation and material flow are acceptable. Forces in the pin were comparable between the welding process and the model as seen in Figure 17. The model predicted all the trends and peaks of forces experienced by the pin within reasonable timing. The forces in the pin from the modeling data, as seen in Figure 17, show three

distinct peaks at t = 0.8 s, t = 0.9 s, and t = 1.1 s. These peaks, respectively, are caused by the material pushing up against the pin at the end of the plunging stage, the pin forced down onto the material in the beginning of the refill stage, and the end of the refill stage when the material is not so easily displaced.

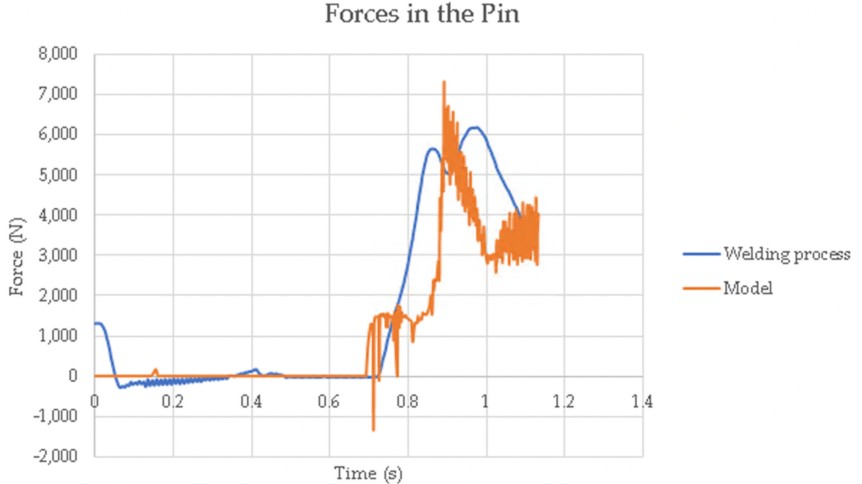

**Figure 17.** Comparison of forces on the pin during the RFFSW process.

Forces in the shoulder, on the other hand, were overpredicted by the model as seen in Figure 18, although the trends were similar. The causes of the overprediction of force experienced in the shoulder tool are likely influenced by the material properties' data of AA 7075-T6. These data for the AA 7075-T6 are composed of two files: one for temperatures below 250 °C and one for temperatures above 250 °C. The validity of these material data files is well-established for modeling metal forming processes, but are perhaps less appropriate for RFSSW, where intense shearing of the material under the tooling is imposed during the process. Future efforts will focus on validating the flow stress data for AA 7075-T6 for FSW/RFFSW, in order to further refine the model predictions. Another possible reason for the overprediction of forces is the rigidity of the clamp. The model uses a rigid displacement boundary condition for the clamp, but in the experiment the clamp is actuated using a gas-powered piston, imposing a target force of 9.84 kN. Although the clamp is not intended to move, it does have some compliance, and thus may relieve stress from the material during the welding process, with corresponding lower forces on the shoulder, compared to the model.

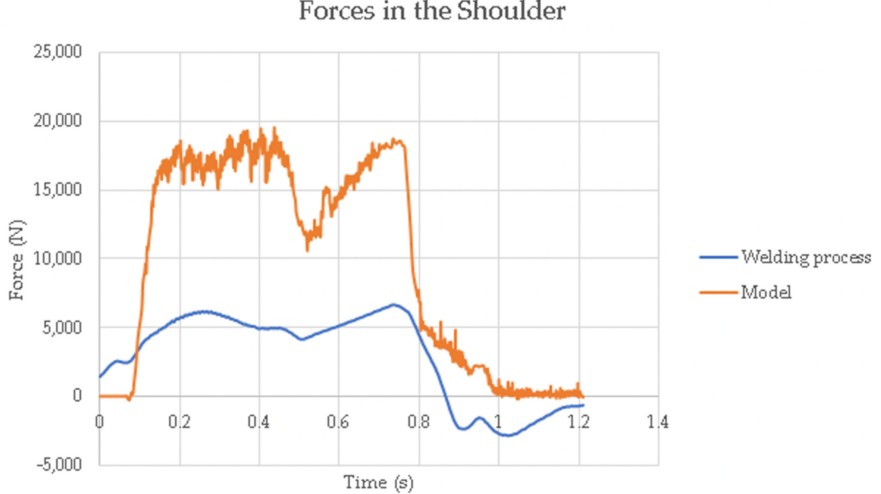

**Figure 18.** Comparison of forces on the shoulder during the RFFSW process.

For the shoulder tool, forces predicted in the model differ from the forces in the welding process by as much as 60%. Although the model overpredicts the forces, the model accurately predicts the trends of the forces of the shoulder tool in the experiment.

## 5. Conclusions

An axi-symmetric 2D model of RFFSW was demonstrated to successfully capture the physics of the welding process as supported by the following conclusions:

1.  The model predicted heat generation from the friction between the tools and the AA 7075-T6 material within 5% of the actual temperatures experienced by the thermocouples in experiments across a range of areas, both inside the weld nugget and in the material directly adjacent to the weld.
2.  The model demonstrated the ability to relate the kinematics of tool motion to material flow in the process by accurately replicating the patterns of defect formation within the weld material. Similarly, with the variation in tool motion, the model demonstrated an ability to predict the reduction/elimination of internal voids, in alignment with experimental observation.
3.  The model has successfully demonstrated its ability to predict the forces experienced by the pin, with progress to be made in its ability to predict the forces experienced by the shoulder. Discrepancies between prediction and experimental forces of the shoulder can likely be attributed to flow stresses for AA 7075-T6 that may be appropriate for the modeling of metal forming, but not for RFSSW, where intense shearing of the material, especially by the shoulder, is not accounted for.

When the force predictions for both the pin and the shoulder are improved with more accurate flow stress data, the model will be of greater utility for the simulation of the RFSSW process under specific parameters. This will allow the estimation of an optimized weld duration, ideally minimizing the duration of the process, while achieving appropriate temperature evolution needed for good bonding.

**Author Contributions:** Formal analysis, E.B. and M.M.; Investigation, A.C. and P.B.; Methodology, Y.H.; Supervision, Y.H.; Validation, A.C.; Writing—original draft, E.B.; Writing—review & editing, M.M. and Y.H. All authors have read and agreed to the published version of the manuscript.

**Funding:** This research was funded by the National Aeronautics and Space Administration (NASA-MSFC) and Boeing Research & Technology through the National Science Foundation IUCRC-Center for Friction Stir Processing (CFSP) (grant number R0602623).

**Conflicts of Interest:** The authors declare no conflict of interest.

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
