# Peer review of "2D Axisymmetric Modeling of Refill Friction Stir Spot Welding and Experimental Validation"

_jmmp, doi:10.3390/jmmp6040089_

Round 1
Reviewer 1 Report
Some figures look a bit blurry. Otherwise, the paper is ready for publication.
The quality of Figure 1 and Figure 4 is not appropriate for publication. Perhaps, it needs higher resolution images.
Author Response
Comment 1:
Some figures look a bit blurry. Otherwise, the paper is ready for publication.
The quality of Figure 1 and Figure 4 is not appropriate for publication. Perhaps, it needs higher resolution images.
Response 1:
Thank you for the feedback. The quality of both figures 1 and 4 were improved and updated in the manuscript with higher resolution images.
Reviewer 2 Report
This paper studied the simulation of Refill Friction Stir Spot Welding. It must be revised before it can be accepted for publication:
Abstract: The main results and conclusions should be taken into account in this section.
According to the 'Materials and Methods' section the authors used in the experiment the tool rotational speed n = 2600 rpm. Why such parameter value was used in investigations?
The values of all physical-mechanical parameters should be presented in the article in order to allow the readers to extend the research results. The reader may not be able to access the Forge database.
Figure 3. I'm surprised that pin motion was higher that thickness of upper sheet.The upper sheet was completely cut by the external edge of sleeve. Penetration of the upper sheet by the pin to a depth less than upper sheet thickness ensures optimal mixing of the materials of both sheets and the highest load capacity. This phenomenon was observed by 99% of authors studied the RFSSW process.
The welding time is also too small for this sheet thickness. The selection of welding parameters should be confirmed experimentally based on the load capacity of the joint. Otherwise, the results are not of practical use.
Authors take into account numerical simulations but did not perform any mesh sensitivity analysis. Such analysis is crucial for the model validation with experimental results. Please add mesh sensitivity study and prove that chosen number of finite elements will not influence on the receiving results.
Description of the numerical simulations should be modify with information consisting type of FE elements.
Fig. 14b, 15b: Scale bars are required.
List of references was not formatted according to the Template.
Drawings showing temperature distribution in the weld zone for specific stags of the RFSSW are welcome.
Author Response
Comment 1:
This paper studied the simulation of Refill Friction Stir Spot Welding. It must be revised before it can be accepted for publication:
Abstract: The main results and conclusions should be taken into account in this section.
According to the 'Materials and Methods' section the authors used in the experiment the tool rotational speed n = 2600 rpm. Why such parameter value was used in investigations?
Response 1:
The abstract was revised to specifically address the conclusions of the paper as related to temperature, material and process forces. Additionally, the methodology was expanded to reference a paper previously published documenting why 2600 RPM was used, and providing more information specific to the plunge depth and volumetric vs. non-volumetric flow.
Comment 2:
The reader may not be able to access the Forge database.
Response 2:
This is an insightful comment. Though we are not able to share access to the Forge database, we can provide the readers with the exact stress/strain curves used for the material properties in the model. This is updated in the manuscript.
Comment 3:
Figure 3. I'm surprised that pin motion was higher that thickness of upper sheet. The upper sheet was completely cut by the external edge of sleeve. Penetration of the upper sheet by the pin to a depth less than upper sheet thickness ensures optimal mixing of the materials of both sheets and the highest load capacity. This phenomenon was observed by 99% of authors studied the RFSSW process.
Response 3:
This is an astute observation, and is correct for the literature from 2011 – 2019. However, in 2020 & 2021 our team previously published results refuting these previous conclusions, and showing that previous conclusions noting this were only accurate for longer process cycle times where diffusion bonding could enable slower bonding within the weld nuggets. I’ve added the references to two of these papers, one of which specifically address weld parameters and plunge depths needed to achieve cycle times below 2 seconds.
Comment 4:
The welding time is also too small for this sheet thickness. The selection of welding parameters should be confirmed experimentally based on the load capacity of the joint. Otherwise, the results are not of practical use.
Response 4:
I believe this comment was largely answered in the response to the previous question. In actually, the most recent studies show that cycle times down to 500 ms can be achieved without any reduction in weld properties. In fact, for precipitation strengthened alloys like 7075, strengths can be increased as the negative influence of heat is reduced for shorter cycle times of the process.
Comment 5:
Authors take into account numerical simulations but did not perform any mesh sensitivity analysis. Such analysis is crucial for the model validation with experimental results. Please add mesh sensitivity study and prove that chosen number of finite elements will not influence on the receiving results.
Response 5:
Three zones of higher mesh density are used in the model, where the mesh has the most re-meshing because of material deformation. During model development, we did refine the mesh progressively until material flow looked reasonable with respect to our experience with experimental cross sections. This has been updated in the manuscript. (p. 7)
Comment 6:
Description of the numerical simulations should be modify with information consisting type of FE elements.
Response 6:
The type of FE elements and the reasoning for their usage is explained in depth on page 6: lines 182-194. These elements have linear shape functions, as denoted by P1+/P1 in the text (both velocity and pressure interpolated using linear shape functions).
Comment 7:
Fig. 14b, 15b: Scale bars are required.
Response 7:
This is a helpful comment. This will add clarity to the comparison of the voids. This has been updated in the manuscript.
Comment 8:
List of references was not formatted according to the Template.
Response 8:
The list of references was updated to the MDPI format according to the template. This is updated in the manuscript. Thank you
Comment 9:
Drawings showing temperature distribution in the weld zone for specific stages of the RFSSW are welcome.
Response 9:
This is a great idea. Images of the model at specific stages have been illustrated to show the temperature distribution throughout the material in the weld zone. This is updated in the manuscript as Figure 14.
Reviewer 3 Report
This paper presents a modeling approach for RFFSW with two tool kinematic modes. The work is validated through experimental measurement of the temperature and forces during the process. Additional details should be provided for the simulation inputs and boundary conditions. Furthermore, the authors should aim to include more discussion rather than pure presentation of the results. Detailed comments to be addressed are below.
· 1) The authors should provide justification for selecting AA-7075 in the introduction. It has many practical applications to industry, but has many challenges in modelling due to the evolving precipitate structure at elevated temperature and strain rates. This may contribute to the discrepancy in shoulder force as described by the authors in Section 4.3.
· 2) On lines 69-70 of the manuscript in the Materials and Methods section, the authors describe the composition and temper. The authors should state whether the composition values are measured or nominal. If measured how were these numbers produced. The statement “30%” increase is not meaningful. A more specific range of yield strength or UTS should be provided, or a general statement about peak aging and precipitate structure should be used.
· 3) All materials properties and constants used in the simulation should be provided either through citation, directly in the text, or in an appendix. In particular, the AA7075 data used to fit Equation 2, and the value and method for determining “q” in Equation 3.
· 4) How is the heat transfer boundary between plates defined in the model. The accuracy of the model is likely sensitive to heat transfer efficiency between these plates.
· 5) For Figure 13 and 14, does the red outline over the semi-transparent area represent a void or does the white spot showing lack of mesh represent a void. In Figure 13, the center of the red outlined region appears to be partially transparent suggesting that the model still predicts a void with smaller size.
· 6) The authors should quantify the differences in force prediction as they did for the temperature prediction.
· 7) The model prediction of the forces in the pin (Fig. 15) has very frequent and steep oscillations. Furthermore, the model appears to have 3 distinct peaks (t = 0.8, 0.9, 1.1) compared to the simulation. What is the cause of this?
Author Response
Comment 1:
This paper presents a modeling approach for RFFSW with two tool kinematic modes. The work is validated through experimental measurement of the temperature and forces during the process. Additional details should be provided for the simulation inputs and boundary conditions. Furthermore, the authors should aim to include more discussion rather than pure presentation of the results. Detailed comments to be addressed are below.
Response 1:
Thanks for these comments. We have added in more detailed discussion related to the overall process, kinematics, comparison, etc. Furthermore, we have addressed more detail in the methodology including additional figures to provide unique details on simulation inputs (especially as related to material properties and boundary conditions)
Comment 2:
The authors should provide justification for selecting AA-7075 in the introduction. It has many practical applications to industry, but has many challenges in modelling due to the evolving precipitate structure at elevated temperature and strain rates. This may contribute to the discrepancy in shoulder force as described by the authors in Section 4.3.
Response 2:
This is an excellent point, and additional detail has been added to the methodology to address the selection of this alloy for the present study.
Comment 3:
On lines 69-70 of the manuscript in the Materials and Methods section, the authors describe the composition and temper. The authors should state whether the composition values are measured or nominal. If measured how were these numbers produced. The statement “30%” increase is not meaningful. A more specific range of yield strength or UTS should be provided, or a general statement about peak aging and precipitate structure should be used.
Response 8:
These values are nominal. This is adjusted in the manuscript, and the “30%” has been erased. UTS and yield strength values are added, and the source is referenced.
Comment 4:
All materials properties and constants used in the simulation should be provided either through citation, directly in the text, or in an appendix. In particular, the AA7075 data used to fit Equation 2, and the value and method for determining “q” in Equation 3.
Response 4:
Representative flow stress curves, for different strains and strain rates, were added to the manuscript for a temperature of 450C, which is typical of the stir zone in RFSSW. The value of q in the friction law is taken as the rate sensitivity of the workpiece material, as the friction law essentially models the shearing of a boundary layer of workpiece material against the tool.
Comment 5:
How is the heat transfer boundary between plates defined in the model. The accuracy of the model is likely sensitive to heat transfer efficiency between these plates.
Response 5:
In this model we have simplified the calculation and modeled the two plates as one mesh in the overlap region. This was done, because the mixing that occurs during RFSSW enhances heat transfer through this zone. In terms of the contact with the backing plate, we have used an adiabatic condition, because the time of welding is very short and not much heat transfer is assumed to occur between lower sheet and backing plate. (pp 7-8)
Comment 6:
For Figure 13 and 14, does the red outline over the semi-transparent area represent a void or does the white spot showing lack of mesh represent a void. In Figure 13, the center of the red outlined region appears to be partially transparent suggesting that the model still predicts a void with smaller size.
Response 6:
Thank you for the comment, this must not have been very clear. The red outline is the boundary where the materials were folded together during welding. These red boundaries represent contact interfaces, where the unilateral contact condition is imposed, rather than allowing the remeshing algorithm to blend the material together. As such, the triangular area with red outline is considered to be the void where material did not bond together. This is updated in the manuscript.
Comment 7:
The authors should quantify the differences in force prediction as they did for the temperature prediction.
Response 7:
Great comment. Quantified differences in force prediction are written as percentages and were added to manuscript.
Comment 8:
The model prediction of the forces in the pin (Fig. 15) has very frequent and steep oscillations. Furthermore, the model appears to have 3 distinct peaks (t = 0.8, 0.9, 1.1) compared to the simulation. What is the cause of this?
Response 8:
Thank you for bringing up such a good point. The first peak at t=0.8 occurs when the material is pushing up against the pin. The second peak at t=0.9 occurs when the pin pushes down onto the material. The third and final peak at t=1.1 occurs when the weld is finishing, and the material is not so easily displaced, therefore the forces against the pin increase. This is updated in the manuscript.
Reviewer 4 Report
The manuscript is well organized. The discussion is quite interesting.
(1) Figure 2 caption
“RFSSW tooling, including a pin (left), shoulder (center), and pin (right)”
“pin (right)” should be “a clamping ring”.
(2) Page 7, line 199
“…, while is the shear stress provided by the friction law…”
It seems that τ is the shear stress.
(3) Figure 14
“Voids were not found in non-volumetric welds with non-symmetric kinematics. Figure 14 shows the mode and experiment, where proper consolidation took place during the joining process.”
Why did proper consolidation take place in non-volumetric welds with non-symmetric kinematics?
Author Response
Comment 1:
(1) Figure 2 caption
“RFSSW tooling, including a pin (left), shoulder (center), and pin (right)”
“pin (right)” should be “a clamping ring”.
Response 1:
Thank you for catching this simple writing error. This is updated in the manuscript.
Comment 2:
(2) Page 7, line 199
“…, while is the shear stress provided by the friction law…”
It seems that τ is the shear stress.
Response 2:
Yes, τ is the shear stress represented in equation 7. This is clear in the manuscript.
Comment 3:
(3) Figure 14
“Voids were not found in non-volumetric welds with non-symmetric kinematics. Figure 14 shows the mode and experiment, where proper consolidation took place during the joining process.”
Why did proper consolidation take place in non-volumetric welds with non-symmetric kinematics?
Response 3:
This is a great comment, as this concept probably was not explained thoroughly enough. Additional discussion in the paper has been added to bring greater clarification to the reader.
Round 2
Reviewer 2 Report
I would like to thank the authors for the effort they have made in replying to all my comments and concerns.
I am satisfied with the reviewed version and the provided explanations to all raised questions.
Author Response
We appreciated your comments, which led to improvements in the manuscript.
Reviewer 3 Report
The authors have addressed the major issues concerning the model input and experimental clarity. The manuscript should be double checked for grammar and spelling mistakes before final submission. Additionally, adding labels denoting the different stages of tool engagement to the figures showing temperature/force values may help to further improve the clarity of presentation.
Author Response
- The authors reviewed the manuscript and corrected remaining grammar and punctuation errors.
- Updates were made to all figures showing temperature and forces as a function of time to distinguish the plunge and refill stages of the process.